# Genome-Wide Identification and Analysis of Family Members with Juvenile Hormone Binding Protein Domains in *Spodoptera frugiperda*

**DOI:** 10.3390/insects15080573

**Published:** 2024-07-28

**Authors:** Yang Liu, Kunliang Zou, Tonghan Wang, Minghui Guan, Haiming Duan, Haibing Yu, Degong Wu, Junli Du

**Affiliations:** 1College of Resources and Environment, Anhui Science and Technology University, Chuzhou 233100, China; 18298115131@163.com (Y.L.); 15665861750@163.com (M.G.); 2College of Agriculture, Anhui Science and Technology University, Chuzhou 233100, China; zkl151231@163.com (K.Z.); 18767401521@163.com (T.W.); duanhm@ahstu.edu.cn (H.D.); yuhb@ahstu.edu.cn (H.Y.)

**Keywords:** fall armyworm, juvenile hormone binding protein, gene family, bioinformatics analysis

## Abstract

**Simple Summary:**

*Spodoptera frugiperda* of the family Noctuidae in the order Lepidoptera, is native to the Americas. It is characterized by its polyphagous nature and high reproductive capacity. *S. frugiperda* has caused significant socio-economic losses in many countries, primarily affecting Poaceae crops such as maize, rice, wheat, and sorghum. This study aims to analyze the members, differentiation, structure, and function of the *JHBP* gene family in *S. frugiperda* through bioinformatics approaches. Subsequently, we utilize real-time quantitative PCR (qPCR) to investigate the expression profiles of this gene family across different developmental stages and various tissues of *S. frugiperda*. The purpose of this analysis is to elucidate the potential roles of these genes, providing key target genes for future functional validation through gene knockdown and RNA interference (RNAi) techniques. This research offers insights that could contribute to the development of new pesticides.

**Abstract:**

Juvenile hormone binding proteins (JHBPs) are carrier proteins that bind to juvenile hormone (JH) to form a complex, which then transports the JH to target organs to regulate insect growth and development. Through bioinformatics analysis, 76 genes encoding *JHBP* in *S. frugiperda* were identified from whole genome data (*SfJHBP1*-*SfJHBP76*). These genes are unevenly distributed across 8 chromosomes, with gene differentiation primarily driven by tandem duplication. Most SfJHBP proteins are acidic, and their secondary structures are mainly composed of α-helices and random coils. Gene structure and conserved motif analyses reveal significant variations in the number of coding sequences (CDS) and a high diversity in amino acid sequences. Phylogenetic analysis classified the genes into four subfamilies, with a notable presence of directly homologous genes between *S. frugiperda* and *S. litura*, suggesting a close relationship between the two species. RNA-seq data from public databases and qPCR of selected *SfJHBP* genes show that *SfJHBP20*, *SfJHBP50*, and *SfJHBP69* are highly expressed at most developmental stages, while *SfJHBP8* and *SfJHBP14* exhibit specific expression during the pupal stage and in the midgut. These findings provide a theoretical basis for future studies on the biological functions of this gene family.

## 1. Introduction

The fall armyworm (*Spodoptera frugiperda*), of the Lepidoptera Noctuidae family, is indigenous to the Americas and known for its omnivorous habits and prolific breeding capabilities [1]. This species displays a broad host range, with a preference for attacking crops such as maize, rice, wheat, sorghum, and other cereals [2]. Noteworthy biological traits of *S. frugiperda*, including its rapid feeding and developmental rates, swift migration patterns, and advantageous hybrid characteristics, enhance its invasive potential. Its adaptability to a wide range of temperatures, host plants, and pesticide resistance facilitates rapid adaptation to new environments, leading to persistent agricultural damage [3]. *S. frugiperda* has inflicted substantial socio-economic losses in many countries, with estimates suggesting annual economic damages reaching billions of dollars in sub-Saharan Africa alone [4]. In 2017, the International Centre for Agricultural and Biological Sciences identified *S. frugiperda* as one of the top ten plant pests worldwide, and its invasion of China in 2019 marked a significant event. Since its incursion [5], the species has been observed annually in extensive areas of China.

JH is a regulatory substance present in insects, crustaceans, and certain plants that is crucial in controlling processes of growth and development [6,7]. During the larval phase, JH orchestrates growth and molting events by modulating specific gene expression patterns that endure into adulthood [8]. JH forms complexes with three categories of JHBPs within the hemolymph. These categories include lipoproteins, hexameric proteins, and low molecular weight proteins with a size of around 30 kDa [9,10]. The role of JHBP, particularly the JH-JHBP complex, in binding to membrane receptors to facilitate transportation has been elucidated, showing that JHBP is a pivotal component in the JH signaling cascade [11,12,13,14]. Additionally, JHBPs safeguard JH from enzymatic degradation by non-specific esterases [15], underscoring their indispensable nature in maintaining the integrity of JH signaling pathways.

The Takeout (TO) protein and JHBP both possess a shared domain that is thought to have the ability to attach to small lipophilic compounds, specifically JH. TO proteins were first discovered and designated in the fruit fly (*Drosophila melanogaster*). JHBPs and TO proteins belong to a superfamily of proteins [16]. TO proteins are prevalent in numerous insect species and play crucial roles in a multitude of physiological functions and behavioral patterns. Saito et al. identified nine *BmJHBP* genes in the silkworm EST database. These *BmJHBP* genes were expressed in various tissues and were regulated by developmental conditions [17]. Dauwalder et al. discovered through in situ hybridization that TO proteins are distributed in the head fat body of male *D. melanogaster*, but are absent in the head fat body of female *D. melanogaster* [18]. TO proteins are highly expressed in the head and abdomen of the green peach aphid (*Myzus persicae*) [19].

JH is an important insect hormone synthesized and secreted by the corpora allata (CA). It exerts unique biological effects within insects through the JHBP present in the hemolymph [20]. JHBP maintains JH at an appropriate titer in the hemolymph by binding or releasing JH, thereby achieving the function of protecting the embryo [21,22]. Insect diapause, depending on the developmental stage, can be classified into four types: embryonic diapause (egg diapause), larval diapause, pupal diapause, and adult diapause. In various studies on insect diapause, JH has been identified as an important hormone that regulates adult diapause, controlling developmental processes and promoting oocyte maturation during the adult stage [20]. The regulatory role of JH in insect growth and development has long been a focus of insect physiology research and has been successfully applied in pest control and beneficial insect utilization [23]. Insect reproductive diapause can be terminated by JH, as evidenced in studies on the black-legged melon leaf beetle (*Aulacophora nigripennis*) [24] and the rice bug (*Scotinophara lurida*) [25]. As the carrier or receptor of JH, JHBP plays a crucial role in the functioning of JH.

The structure and expression patterns of the *JHBP* gene have been elucidated in the tobacco hornworm (*Manduca sexta*) [26]. Xiuting He et al. cloned the *JHBP* gene from the small brown planthopper (*Laodelphax striatellus*), measuring gene expression across different larval stages. They discovered that the levels of gene expression were notably lower in all stages compared to the fourth instar larvae, and there was no distinction noted between the adult males and females [27]. Previous research has identified that the *OfJHBP* gene within the bamboo borer (*Omphisa fuscidentalis*) has the ability to impact larval diapause and serves as a key regulator in the processes of growth, development, and diapause [28]. In the greater wax moth (*Galleria mellonella*), *JHBP* mRNA levels peak during the final larval stage and decrease fivefold after pupation, indicating that *JHBP* can influence pupation [29]. Long Chen et al. cloned the *JHBP* gene from the Daursky beetle (*Galeruca daurica*), finding that *GdJHBP* is present in the hemolymph and expressed at various developmental stages, with higher expression before and after adult diapause, suggesting a significant role in adult summer diapause [30]. In the cotton bollworm (*Helicoverpa armigera*), *HaJHBP* expression is significantly suppressed under starvation conditions, leading to delayed larval development [31]. These studies collectively suggest that *JHBP* participates in regulating various stages of insect growth and development, and can also impact feeding, reproduction, mating, drug resistance, and immune responses [32,33].

This study investigated the bioinformatics of the *SfJHBP* gene family and its expression patterns in various developmental stages of *S. frugiperda*, as well as in different tissues of fifth instar larvae. The research analyzed the potential role of the gene and validated its function for future gene knockdown experiments and the application of RNAi technology. Understanding the control of *S. frugiperda* is crucial for identifying key target genes and guiding the development of new pesticides.

## 2. Materials and Methods

### 2.1. Identification and Chromosomal Localization of the JHBP Gene Family in S. frugiperda

Focusing on *S. frugiperda* as the research subject, the genomic sequence information file and annotation information file of the Zhejiang University version of *S. frugiperda* were obtained from the NCBI website (https://www.ncbi.nlm.nih.gov/ accessed on 12 February 2024). The HMM model file (PF06585) for the *JHBP* gene family was downloaded from the Pfam (version 37.0) database (http://pfam.xfam.org/ accessed on 15 February 2024) [34]. Using the software Tbtools (version 2.096) [35], the protein sequences of *S. frugiperda* were extracted and duplicate sequences were removed. These protein sequences were then examined for the presence of JHBP domains using tools such as HMMER (Version 3.0) [36], NCBI’s Conserved Domain Search (https://www.ncbi.nlm.nih.gov/Structure/cdd/wrpsb.cgi accessed on 17 February 2024) [36], and SMART (Version 9.0) (SMART: Main page (http://smart.embl-heidelberg.de/) accessed on 17 February 2024) [37], excluding those without JHBP domains.

The physicochemical properties of the *JHBP* gene family in *S. frugiperda* were analyzed using ExPaSy (https://web.expasy.org/protparam/ accessed on 19 February 2024) [38]. Subcellular localization was predicted using WOLFPSORT (https://wolfpsort.hgc.jp/ accessed on 22 February 2024) [39]. Secondary structures of the proteins were predicted using SOPMA (https://npsa-prabi.ibcp.fr/cgi-bin/npsa_automat.pl?page=npsa_sopma.html accessed on 25 February 2024) [40]. Finally, the chromosomal distribution map of the JHBP gene family in *S. frugiperda* was created using TBtools (version 2.096) [35].

### 2.2. SfJHBP Gene Duplication and Ka/Ks Value Analysis

The software TBtools (version 2.096) [35] was used to analyze the duplication relationships among genes within the species and to calculate the non-synonymous substitution rate (Ka) and synonymous substitution rate (Ks). A Ka/Ks ratio greater than 1, less than 1, or equal to 1 indicates positive, negative, or neutral evolution, respectively [41]. The divergence time was calculated using the formula T = Ks/(2λ) (λ = 6.5 × 10^−9^) [42].

### 2.3. Structure and Conserved Motif Analysis of SfJHBP Genes

Motif analysis of *S. frugiperda* proteins was conducted using MEME (Version 5.5.5) (https://meme-suite.org/meme/tools/meme accessed on 3 March 2024) [43], with the number of motifs set to 10 and other parameters set to default. The gene structure and conserved motifs of the JHBP gene family in *S. frugiperda*, along with their phylogenetic tree, were integrated using TBtools (version 2.096) [35].

### 2.4. Phylogenetic and Synteny Analysis of the SfJHBP Gene Family

To investigate the evolutionary relationships between *S. frugiperda* and other species, genomic sequence and annotation files for *Spodoptera litura* Fabricius (https://www.ncbi.nlm.nih.gov/datasets/genome/GCF_002706865.2/ accessed on 7 March 2024) and *Plutella xylostella* Linnaeus (https://www.ncbi.nlm.nih.gov/datasets/genome/GCF_932276165.1/ accessed on 10 March 2024) were downloaded from the NCBI database. Using the HMM model file (PF06585) [34], JHBP protein sequences were extracted using TBtools (version 2.096) [35], and duplicates and sequences lacking JHBP domains were removed. The homologous JHBP protein sequences were then used to construct a phylogenetic tree in MEGA (Version 11.0.13) [44], and the tree was visualized using iTOL (Version 6.8.2): (https://itol.embl.de/ accessed on 20 March 2024) [45]. Synteny analysis was performed using TBtools (version 2.096) [35], based on gene location files on chromosomes and interspecies gene association files.

### 2.5. Analysis of Expression Patterns of the SfJHBP Gene Family in Diverse Developmental Stages and Tissues of S. frugiperda

Transcripts of the JHBP gene family are detected in various developmental stages of *S. frugiperda* including 1–6 instars, pupae, and both male and female adult stages, as well as different tissues such as midgut, head, cuticle, fat body, hemolymph, and malpighian tubule. The data set utilized in this study was obtained from the NCBI’s SRA database (https://www.ncbi.nlm.nih.gov/sra/?term= accessed on 26 March 2024), and the accession numbers are PRJNA590312 [46] and PRJNA1070356 [47] respectively; the data were processed using HeatMap of TBtools (version 2.096) [35] to obtain the SfJHBP gene family expression content heat map.

### 2.6. qPCR Expression Characteristics of Some SfJHBP Genes in Different Developmental Stages and Different Tissues of S. frugiperda

#### Insect Rearing

Larvae of *S. frugiperda* were gathered from the maize experimental field at Anhui Science and Technology University and then nurtured in a controlled climate chamber. They were fed fresh maize leaves within rearing containers, with adults receiving 10% honey water. Following several generations of rearing indoors, the insects were utilized for experiments. The conditions of the artificial climate chamber were set as follows: temperature (25 ± 1) °C, relative humidity (70 ± 5)%, and photoperiod 16:8 (L) hours.

Samples of *S. frugiperda* eggs (*n* = 100), 1st instar (*n* = 30), 2nd-6th instar larvae (*n* = 10), pupae (*n* = 5), male and female adults (*n* = 5), and different tissues of 5th instar larvae (fat body, Malpighian tubules, integument, midgut, head, hemolymph) (*n* = 15) were collected. Each sample was replicated three times, flash-frozen in liquid nitrogen, and stored at −80 °C for RNA extraction.

RNA extraction and reverse transcription we’re performed according to Lv et al. [48]. Primers for each specific gene were designed using Primer 3.0 [49], with GADPH as the reference gene (Table 1). qPCR was conducted according to Jin et al. [50], measuring the relative expression levels of 13 SfJHBP genes across various developmental stages and tissues(From the transcriptome data of various developmental phases and tissues of S. frugiperda, we selected 13 representative highly expressed genes). Each sample was tested in triplicate. The selected genes’ expression levels were calculated using the 2^−ΔΔCT^ method [51], then analyzed with SPSS software (Version 28) [52]. Data visualization was done using GraphPad Prism (Version 9.5.0) [53].

## 3. Results

### 3.1. Identification of SfJHBP Gene Family and Chromosomal Localization Analysis

The genes discovered in this study have been given the names SfJHBP1 to SfJHBP76 in order of their placement on the chromosomes (Figure 1). The analysis revealed that the 76 *JHBP* genes exhibited an irregular distribution pattern. Chromosome 4 exhibited the largest quantity of genes (totaling 37). This was succeeded by chromosome 22, which contained 31 genes. Additionally, chromosomes 2 and 5 each had two genes, while chromosomes 7, 20, 23, and 25 each harbored a single gene. Notably, the *SfJHBP* genes on chromosomes 4 and 22 predominantly occurred in gene clusters.

### 3.2. SfJHBP Protein Structure Analysis

#### 3.2.1. Examination of the Physicochemical Characteristics and Subcellular Localization of the Proteins

By analyzing the physical and chemical properties of the protein, it was found that its amino acid length ranges from 189 aa to 544 aa, with a large length difference, and its molecular weight ranges from 21,806.51 Da to 60,148.08 Da. The amino acid length is proportional to its molecular weight; there are 66 SfJHBP proteins The sequence instability coefficient is less than 40, which means it is a stable protein. By analyzing the overall average hydrophobicity, 47 SfJHBP proteins have negative hydrophobicity. It is speculated that these 47 are hydrophilic proteins and the remaining 29 are hydrophobic proteins (Appendix A).

Subcellular localization analysis identified 60 JHBP proteins in the extracellular region, two in the plasma membrane, eight in the endoplasmic reticulum, four in the cytoplasm, and two in the mitochondria. While the SfJHBP proteins exhibited diverse subcellular distributions, most were localized to the extracellular region, suggesting a primary functional role outside the cells.

Isoelectric point analysis revealed an isoelectric point below 7 for SfJHBP1-SfJHBP54, indicating an acidic nature, and above 7 for SfJHBP55-SfJHBP76, classified as alkaline proteins. This finding indicated that most members of this family were acidic.

#### 3.2.2. Secondary Protein Structure

The structure of a protein molecule determines its function. As seen from the proportions listed in Appendix A, the dominant secondary structure elements are α-helix > random coil > extended strand > β-turn. However, there are 17 proteins (SfJHBP24, SfJHBP31, SfJHBP32, SfJHBP35, SfJHBP36, SfJHBP42, SfJHBP46, SfJHBP49, SfJHBP55, SfJHBP62, SfJHBP64, SfJHBP67, SfJHBP68, SfJHBP70, SfJHBP73, SfJHBP74, SfJHBP76) where the random coil structure predominates. The proportion of β-turn structure is generally low, usually below 10%. It is hypothesized that α-helix and random coil structures may play dominant roles in the secondary structure of proteins, the extended strand structure might serve a secondary role, and the β-turn structure is likely to have a certain modifying function.

Nine SfJHBP protein pairs exhibited identical secondary structure ratios (SfJHBP3 and SfJHBP9; SfJHBP4 and SfJHBP10; SfJHBP5 and SfJHBP11; SfJHBP6 and SfJHBP12; SfJHBP7 and SfJHBP13; SfJHBP8 and SfJHBP14; SfJHBP23 and SfJHBP28; SfJHBP32 and SfJHBP36; SfJHBP73 and SfJHBP74; Appendix A), indicating that two genes might encode the same protein.

### 3.3. Examination of SfJHBP Gene Duplication and Ka/Ks Ratios

A comprehensive analysis of the evolutionary relationships and chromosomal localization of family members identified 29 gene pairs involved in tandem replication events, as detailed in Appendix A. This finding underscores the tandem replication mechanisms through which the *JHBP* gene family expansion predominantly occurred. Gene divergence occurred approximately 10.92–84.62 million years ago. Among the replicators, 20 exhibited Ka values smaller than Ks, while only one displayed a Ka value larger than Ks, suggesting a prevalent synonymous substitution pattern. Notably, the Ka/Ks ratio was less than 1.0 for 19 gene pairs, indicating intense purifying selection and minimal functional differentiation. Conversely, the Ka/Ks ratio exceeded 1.0 in two gene pairs, suggesting significant positive selection and pronounced functional divergence. Notably, eight gene pairs exhibited a Ka/Ks ratio that indicated substantial sequence divergence and a considerable evolutionary distance (NaN values detected).

### 3.4. Structure and Conserved Motif Analysis of SfJHBP Gene

Analysis of the structural distribution diagrams for the 76 JHBP gene introns and UTR regions demonstrated variations in both number and length (Figure 2). Notably, *SfJHBP19* had three CDSs without UTR regions, *SfJHBP17*, *SfJHBP55*, and *SfJHBP64* had five, and *SfJHBP38* had six.

The analysis found that the JHBP proteins of *S. frugiperda* have a conserved motif with significant length variations, illustrating the relatively high diversity among *JHBP* family gene members. The distribution of these conserved motifs in *S. frugiperda* JHBP proteins was further analyzed (Figure 2). *SfJHBP54*, *SfJHBP73* and *SfJHBP74* all contain only one conserved motif.

### 3.5. Cross-Species Evolutionary Analysis

The comprehensive NCBI database search identified 169 homolog sequences of *JHBP*, 76 from *S. frugiperda*, 40 from *P. xylostella*, and 53 from *S. litura*. The constructed phylogenetic tree (Figure 3) delineated four distinct groups: Group A, characterized by the lowest number of *JHBP* genes (two genes, one from *S. frugiperda* and one from *S. litura*); Group B, containing 46 genes (26 from *S. frugiperda*, six from *P. xylostella*, and 14 from *S. litura*); Group C, consisting of nine genes (five from *S. frugiperda*, one from *P. xylostella*, and three from *S. litura*); Group D was the largest group with 112 genes (44 from *S. frugiperda*, 33 from *P. xylostella*, and 35 from *S. litura*).

This research identified a collective of 33 pairs of genes that are orthologous across the different species, including 30 pairs between *S. frugiperda* and *S. litura*, 2 pairs between *S. litura* and *P. xylostella*, and 1 pair between *S. frugiperda* and *P. xylostella*; *S. frugiperda* and *S. litura* were more homologous genes with closer relationship, while *S. frugiperda* and *P. xylostella* were the least homologous genes with more distant relationship.

### 3.6. Cross-Species Covariance Analysis

The analysis revealed that *S. frugiperda* and *P. xylostella* exhibited 12 co-linear segments harboring *JHBP* genes, encompassing 23 gene pairs (Figure 4). *S. frugiperda* and *S. litura* displayed 14 co-linear segments with *JHBP* genes, comprising 36 gene pairs. These findings indicated a higher level of genomic homology between *S. frugiperda* and *S. litura* than between *S. frugiperda* and *P. xylostella*.

### 3.7. SfJHBP Gene Family Expression at Various Developmental Stages

The data analysis indicated that *SfJHBP60* and *SfJHBP69* exhibited high expression levels across developmental stages (Figure 5), while six genes (*SfJHBP17*, *SfJHBP24*, *SfJHBP29*, *SfJHBP33*, *SfJHBP37*, and *SfJHBP38*) were poorly expressed in all developmental stages. Additionally, 27 genes (*SfJHBP3*, *SfJHBP4*, *SfJHBP6*, *SfJHBP7*, *SfJHBP8*, *SfJHBP9*, *SfJHBP10*, *SfJHBP12*, *SfJHBP13*, *SfJHBP14*, *SfJHBP18*, *SfJHBP19*, *SfJHBP20*, *SfJHBP25*, *SfJHBP30*, *SfJHBP34*, *SfJHBP39*, *SfJHBP40*, *SfJHBP44*, *SfJHBP45*, *SfJHBP47*, *SfJHBP50*, *SfJHBP63*, *SfJHBP66*, *SfJHBP69*, *SfJHBP70*, and *SfJHBP76*) were highly expressed in 1st to 5th instar larvae of *S. frugiperda*, 13 (*SfJHBP18*, *SfJHBP19*, *SfJHBP21*, *SfJHBP26*, *SfJHBP43*, *SfJHBP44*, *SfJHBP45*, *SfJHBP50*, *SfJHBP60*, *SfJHBP73*, *SfJHBP74*, and *SfJHBP76*) were highly expressed in the sixth instar larvae, pupae, and adult females and males.

### 3.8. Expression Analysis of SfJHBP Gene Family in Different Tissues of S. frugiperda

As the findings, depicted in Figure 6, show, 13 genes (*SfJHBP3*, *SfJHBP4*, *SfJHBP6*, *SfJHBP7*, *SfJHBP8*, *SfJHBP9*, *SfJHBP10*, *SfJHBP12*, *SfJHBP13*, *SfJHBP14*, *SfJHBP30*, *SfJHBP34*, and *SfJHBP39*) were highly expressed in the midgut, 19 (*SfJHBP1*, *SfJHBP2*, *SfJHBP20*, *SfJHBP25*, *SfJHBP40*, *SfJHBP41*, *SfJHBP42*, *SfJHBP44*, *SfJHBP46*, *SfJHBP47*, *SfJHBP50*, *SfJHBP51*, *SfJHBP60*, *SfJHBP62*, *SfJHBP66*, *SfJHBP69*, *SfJHBP70*, *SfJHBP75*, and *SfJHBP76*) were highly expressed in the head and cuticle tissues, six (*SfJHBP18*, *SfJHBP22*, *SfJHBP27*, *SfJHBP60*, *SfJHBP61*, and *SfJHBP71*) were highly expressed in fat tissue, three (*SfJHBP62*, *SfJHBP73*, and *SfJHBP74*) were highly expressed in the hemolymph, and four (*SfJHBP33*, *SfJHBP37*, *SfJHBP39*, and *SfJHBP71*) were highly expressed in the Malpighian tubules.

### 3.9. qPCR Expression Analysis of Some SfJHBP Genes in Different Developmental Stages and Different Tissues of S. frugiperda

In the developmental stages of *S. frugiperda*, expression levels of 13 *SfJHBP* genes showed notable differences (Figure 7). Notably, *SfJHBP8* and *SfJHBP14* exhibited similar expression patterns, with peak expression during the pupa stage and minimal or absent expression in other phases. Additionally, the levels of *SfJHBP18*, *SfJHBP19*, *SfJHBP20*, *SfJHBP40*, *SfJHBP50*, *SfJHBP66*, *SfJHBP69*, and *SfJHBP76* were elevated in the fifth instar compared to other stages. *SfJHBP47* and *SfJHBP60* had higher expression in the fourth instar than in different phases. *SfJHBP25* had the highest expression during the first instar, followed by the fourth and fifth instars.

In summary, among these 13 genes, 8 *SfJHBP* genes are highly expressed in the fifth instar stage, followed by 2 genes in the fourth instar stage, while *SfJHBP8* and *SfJHBP14* are only specifically expressed in the pupal stage.

The levels of expression of 13 *SfJHBP* genes differ significantly among various tissues of *S. frugiperda* (Figure 8). Among these genes, *SfJHBP8* and *SfJHBP14* have similar expression patterns, with high levels only in the midgut and low levels in other tissues. On the other hand, *SfJHBP18* shows the highest expression in the head, followed by the Malpighian tubules, with no expression in the hemolymph. *SfJHBP20* and *SfJHBP25* exhibit comparable expression patterns, with the head having the highest expression levels followed by the integument. In contrast, *SfJHBP40*, *SfJHBP47*, *SfJHBP50*, *SfJHBP66*, *SfJHBP69*, and *SfJHBP76* display similar expression patterns, with the integument having the highest expression levels followed by the head. *SfJHBP19* is expressed in all tissues, with high levels in the head, integument, and fat body, while *SfJHBP60* shows high expression in the fat body followed by the head.

In summary, among these 13 genes, 6 *SfJHBP* genes are highly expressed in the integument, 4 are highly expressed in the head, *SfJHBP8* and *SfJHBP14* are specifically expressed in the midgut, and *SfJHBP60* is specifically expressed in the fat body.

## 4. Discussion

This study focuses on the *JHBP* gene family in *S. frugiperda*. We identified the members of this gene family and conducted analyses on their physicochemical properties, gene structure, subcellular localization, protein secondary structure, sequence characteristics, phylogenetic relationships, chromosomal localization, interspecies synteny, and gene expression. The analysis of *JHBP* gene family fills the gap of *JHBP* related research in *S. frugiperda* to some extent.

To date, some *JHBP* gene families have been identified in species such as the cypress looper (*Dendrolimus houi Lajonquiere*) [54] and the silkworm (*Bombyx mori*) [55] However, a comprehensive identification of *JHBP* genes in the genome of *S. frugiperda* has not been reported. In this study, 76 *JHBP* gene family members were identified at the whole-genome level in *S. frugiperda* (Appendix A. *SfJHBP* genes are distributed across 8 chromosomes, and the number of *JHBP* family members varies among different species: 47 in *D. houi* [54] and 41 in *B. mori* [55]. This variation in the number of family members across species is likely a consequence of evolutionary processes, where gene family expansion or contraction occurs as an adaptive response to differing environmental conditions over long-term evolution.

Gene duplication is a critical mechanism for the evolution of genomes and species. This process is essential for the functional divergence of homologous genes and is a major factor in the development of new functional genes [56]. The different modes of gene duplication, such as segmental duplication, whole-genome duplication, tandem duplication, and transpositional duplication, play unique roles in shaping genetic diversity and complexity [57]. Through gene duplication, organisms can acquire novel traits and adapt to changing environments. This process not only contributes to the diversity of gene function but also plays a vital role in evolutionary processes. By understanding the various modes of gene duplication and their implications, valuable insights can be gained into the mechanisms driving genetic evolution. This study found that among the 76 *SfJHBP* genes, 29 gene duplication events occurred, all of which were tandem duplications, indicating that the expansion of *SfJHBP* genes was achieved through tandem duplication.

JH plays a pivotal role in regulating numerous essential life events in insects. Within the Lepidoptera order, which includes significant agricultural pests, the signaling of JH is tightly governed by species-specific JHBPs in the hemolymph. These JHBPs, characterized by their high-affinity and low molecular weight, are responsible for transporting JH from its site of synthesis to the target tissues. Hence, JHBPs show great potential as targets for creating innovative insect growth regulators (IGRs) and insecticides. JH are acyclic sesquiterpenoids that consist of α, β-unsaturated ester and terpene skeleton with a terminal epoxide. The hormone’s regulatory functions are dependent on both the ester and epoxide groups. JH plays a crucial role in controlling diverse processes like insect growth, development, and reproduction [6,7]. Through the analysis of *SfJHBP* gene family expression levels at various developmental stages, combining transcriptome data and qPCR analysis, revealed that *SfJHBP20*, *SfJHBP50* and *SfJHBP69* are significantly expressed in multiple developmental stages, indicating their pivotal role in growth and development.

Previous studies have shown that the expression levels of JHBPs in insects are consistent with changes in JH titer [58]. It is thus inferred that JHBPs in *S. frugiperda* exhibit similar expression patterns. Combining RNA-seq data from public databases and qPCR analysis of specific *SfJHBP* genes at different developmental stages, we found that *SfJHBP18*, *SfJHBP19*, *SfJHBP20*, *SfJHBP25*, *SfJHBP40*, *SfJHBP50*, *SfJHBP66*, and *SfJHBP76* are most highly expressed at the 5th instar and decrease as the larvae progress to the 6th instar and pupate. This pattern reflects *S. frugiperda*’s varying needs for JH. As the larvae enter the 5th instar, their food intake significantly increases, and the expression of JHBPs rises, possibly regulating feeding behavior [17]; the significant reduction or absence of JH titer at the late 6th instar promotes molting and pupation [59], possibly related to internal and external morphological and structural reconstruction before eclosion [60]. This trend is similar to the declining expression of *OfJHBP* mRNA in *O. fuscidentalis* from the diapause to pupation stage, indicating JHBP’s crucial role before diapause pupation and eclosion. Furthermore, interference with JHBP in unmated female brown planthoppers (*Nilaparvata lugens*) significantly reduces their mating desire [61], suggesting a regulatory role of *JHBP* in adult reproductive behavior. In this study, the increased expression of *SfJHBP18*, *SfJHBP19*, *SfJHBP20*, *SfJHBP40*, *SfJHBP47*, and *SfJHBP76* in adult insects may also be related to reproductive regulation, which requires further verification.

JH is vital for maintaining insect growth and morphology. A reduction in JH levels before molting can result in incomplete pupation [8]. Combining transcriptome data from different tissues and qPCR analysis of specific *SfJHBP* genes at various developmental stages revealed that the primary action sites of the *SfJHBP* gene family are the midgut, head, and integument tissues of *S. frugiperda*. The expression sites of JHBPs vary among different insects. For instance, *GdJHBP* is highly expressed in the thorax and abdomen of adult cypress looper beetles but low in the head [30]. In *B.mori*, *JHBP* is mainly expressed in the fat body of 4th instar larvae, with minor expression in the midgut, silk gland, and testis [17]. In the Italian honeybee (*Apis mellifera*), *JHBP* expression is detected only in the head and abdomen but not in other parts [62]. The cytoplasmic *JHBP* in the Chinese oak silkworm (*Antheraea pernyi*) is highly expressed in the Malpighian tubules and variably expressed in the hemolymph, fat body, silk gland, epidermis, testis, ovary, brain, and muscles [59]. The *JHBP* in *O*. *fuscidentalis* is highly expressed in the fat body of 5th instar larvae and lowly expressed in the brain, prothoracic gland, epidermis, gut, and silk gland [28]. These differences may be due to species variation, developmental stages, physiological conditions, or *JHBP* types.

The expression characteristics of the *SfJHBP* gene family in different developmental stages and tissues of *S. frugiperda* provide insights into the potential functions of specific genes. This information lays a theoretical foundation for future gene knockdown experiments to verify their functions and their efficacy in controlling *S. frugiperda*, offering references for developing new pesticides.

## 5. Conclusions

A total of 76 *JHBP* genes were identified from the genome of *S. frugiperda*. These genes are unevenly distributed across eight chromosomes. The differentiation of these genes primarily occurred through tandem duplication. The secondary structure of the proteins is predominantly composed of α-helices and random coils. There are a considerable number of orthologous genes shared between *S. frugiperda* and *S. litura*, indicating a close evolutionary relationship. Through qPCR analysis of 13 *SfJHBP* genes, it was found that *SfJHBP20*, *SfJHBP50*, and *SfJHBP69* exhibit high expression levels at multiple developmental stages. Additionally, *SfJHBP40*, *SfJHBP50*, *SfJHBP66*, *SfJHBP69*, and *SfJHBP76* are highly expressed in both the fifth instar larvae and the integument. *SfJHBP8* and *SfJHBP14* are specifically expressed during the pupal stage and in the midgut, while *SfJHBP60* shows specific expression in the fat body. This study provides a basis for predicting the functions of *JHBP* genes in *S. frugiperda*, offering theoretical support for subsequent gene knockout experiments to validate their roles. Furthermore, the findings could inform strategies for controlling *S. frugiperda* and contribute to the development of new pesticides.

## Figures and Tables

**Figure 1 insects-15-00573-f001:**
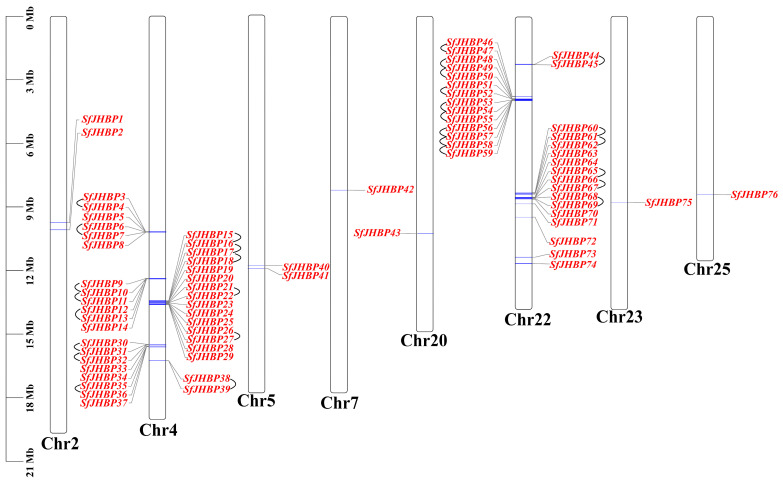
Chromosomal localization of *SfJHBP* gene. Note: 76 *SfJHBP* genes are distributed on the 8 chromosomes of *S. frugiperda*. Chr represents the chromosome, and the genes connected by black lines represent gene pairs that are tandemly replicated on the same chromosome.

**Figure 2 insects-15-00573-f002:**
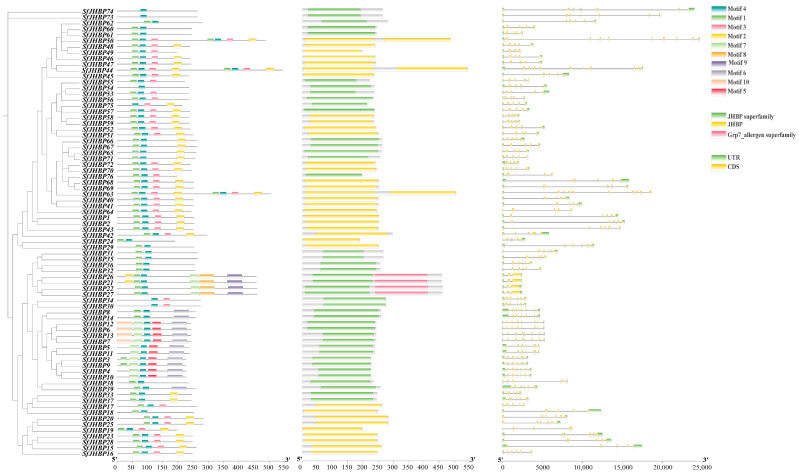
Presents a schematic depiction of conserved motifs, domains, and the exon-intron architectures within the SfJHBP protein of *S. frugiperda*.

**Figure 3 insects-15-00573-f003:**
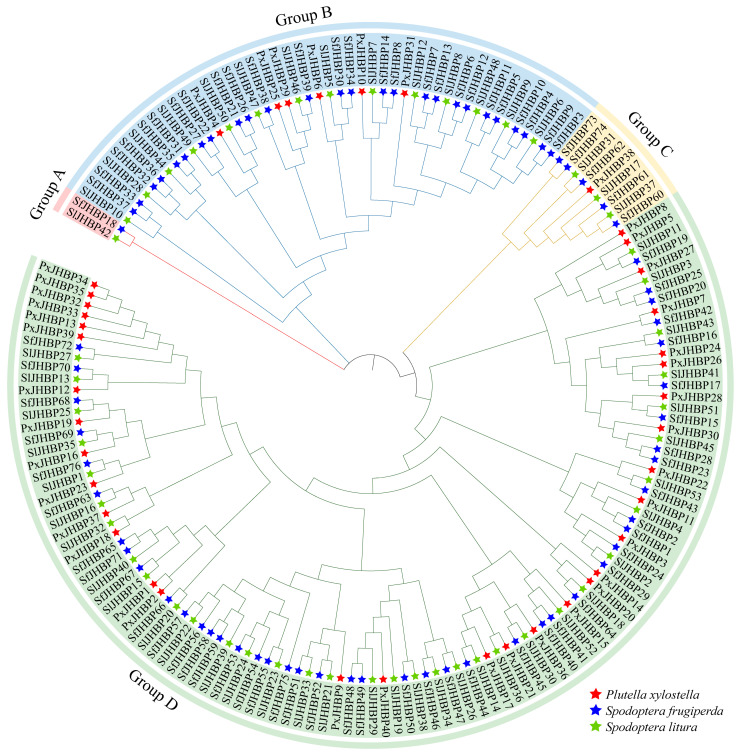
Phylogenetic tree analysis of JHBP protein in *S. frugiperda*, *S. litura* and *P. xylostella*.

**Figure 4 insects-15-00573-f004:**
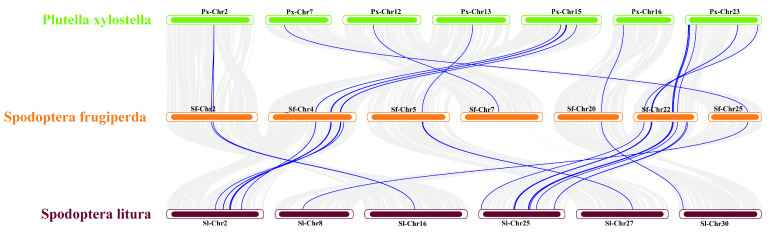
Collinearity analysis of *JHBP* genes of *S. frugiperda*, *S. litura* and *P. xylostella.* Note: The figure shows the distribution of homologous genes in different species, and the blue line represents homologous genes between species.

**Figure 5 insects-15-00573-f005:**
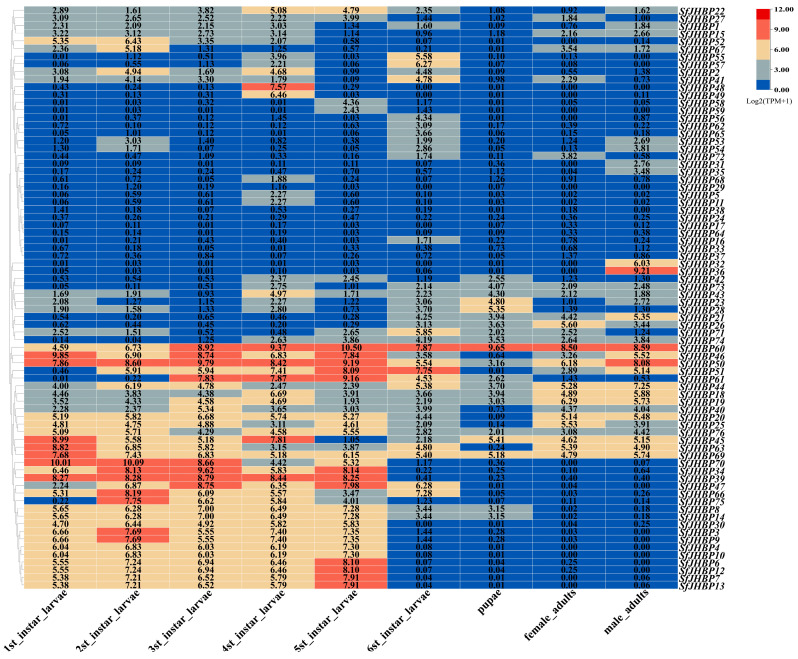
Expression analysis of *SfJHBP* gene family in different developmental stage of *S. frugiperda*.

**Figure 6 insects-15-00573-f006:**
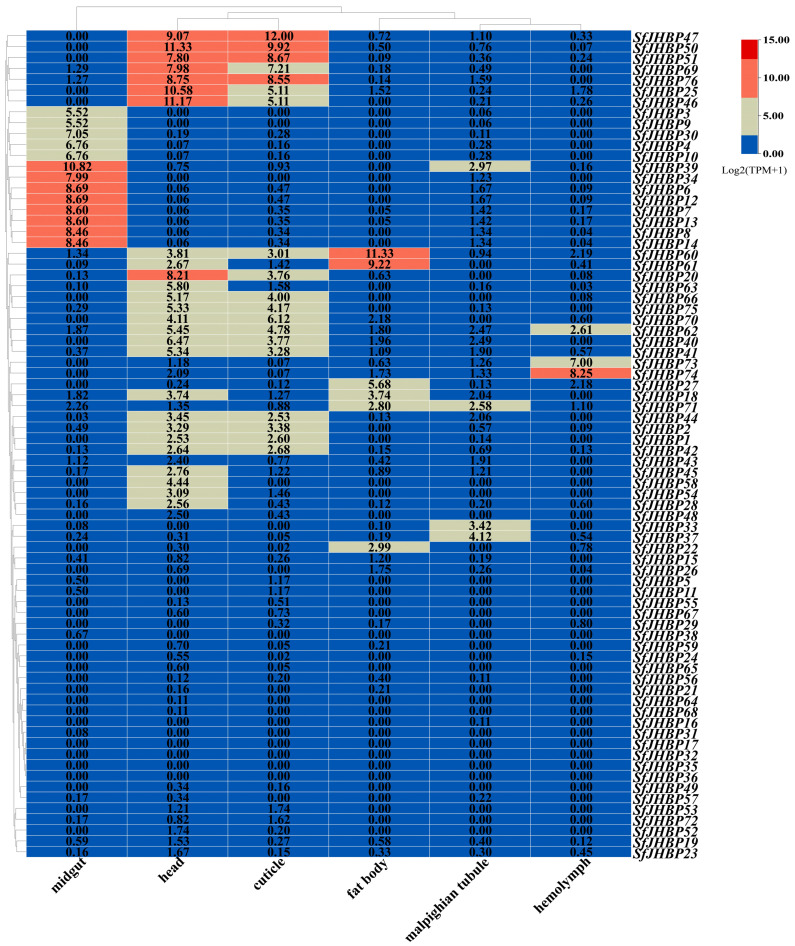
Expression analysis of *SfJHBP* gene family in different organs of *S. frugiperda*.

**Figure 7 insects-15-00573-f007:**
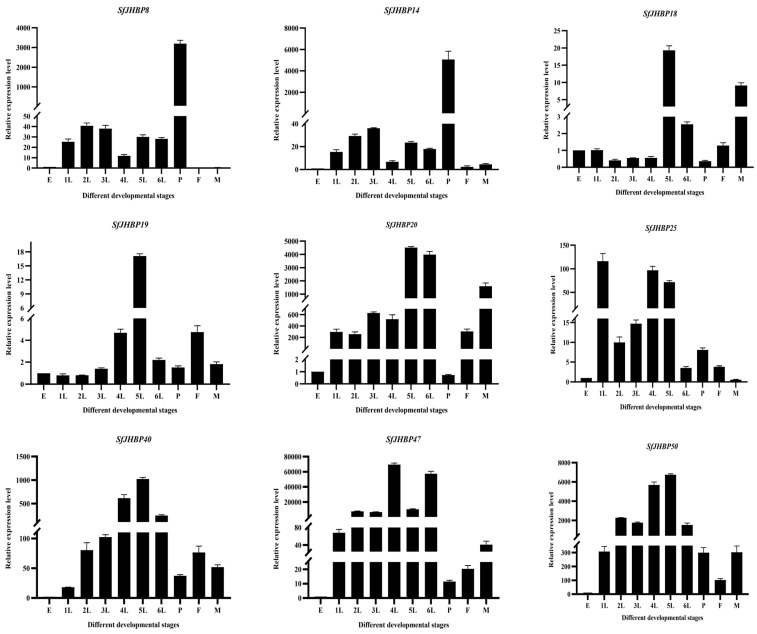
Expression patterns of 13 *SfJHBP* genes at different developmental stages of *S. frugiperda*. Note: 1L: 1st instar 2L: 2nd instar 3L: 3rd instar 4L: 4th instar 5L: 5th instar 6L: 6th instar P: pupa stage F: female adult stage M: male adult stage Data points represent average Value ± standard deviation. In the expression of *13 SfJHBP* genes in different developmental stages of *S. frugiperda*, we used the egg stage as the control group.

**Figure 8 insects-15-00573-f008:**
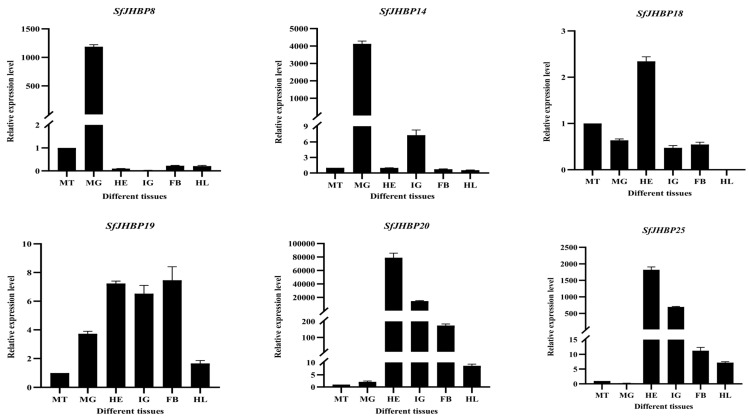
Expression patterns of 13 *SfJHBP* genes in different tissues of *S. frugiperda*. Note: MT: Malpighian tube MG: midgut HE: head IG: integument FB: fat body HL: hemolymph. For the expression of 13 *SfJHBP* genes in different tissues of *S. frugiperda*, we used Malpighian tubules as the control group.Data points represent mean ± standard deviation.

**Table 1 insects-15-00573-t001:** Sequence of primers in this experiment.

Primer Name	Sequence of Primers (5′-3′)
*GADPH-F*	CGGTGTCTTCACAACCACAG
*GADPH-R*	TTGACACCAACGACGAACAT
*SfJHBP8-F*	CTGCTGAGGCTAAGAACTACGA
*SfJHBP8-R*	CATTACTGGTTGTGATGCTGTAGA
*SfJHBP14-F*	ACGCCAGTGGATGTCTCAA
*SfJHBP14-R*	ATCATTAGAGCCTGTCACCATATC
*SfJHBP18-F*	ACTGAAGAGTCCAAAGCTAGATTG
*SfJHBP18-R*	ACCTTACAAACGACGTTGATGA
*SfJHBP19-F*	TCTTCCATCTACCTGACCAACTT
*SfJHBP19-R*	GTCCGCAAGGCTCTTCTCTA
*SfJHBP20-F*	CTGAACGCTGATGCTGTGAA
*SfJHBP20-R*	CTCTGTAGTCCTTGATGCTGATG
*SfJHBP25-F*	GCGTCGTGAAGTCCATAGAAG
*SfJHBP25-R*	ATGAGCGTGTTGTCTGATGTC
*SfJHBP40-F*	AGATTACTTCGCAACCAGCATT
*SfJHBP40-R*	GTACGCCAGTAGAAGAATCACAA
*SfJHBP47-F*	AGTAACTTGTTCAACGGTGACA
*SfJHBP47-R*	AATGCTTGTGACGCCTTCC
*SfJHBP50-F*	TGCTCAGAACGGTAATGATGTG
*SfJHBP50-R*	TTATTGGTACTGCGAGGAAGAAG
*SfJHBP60-F*	GGAGAACCAGTTATATCGCTTGA
*SfJHBP60-R*	AAGGCACGGCACGGATAA
*SfJHBP66-F*	TGACTTCACGGACGATGAGT
*SfJHBP66-R*	TGCCATTGTCTGTTGCTTCTT
*SfJHBP69-F*	ATGTCGGCGGTTGAAGGT
*SfJHBP69-R*	GGCGGTTGTACTACTGTATCG
*SfJHBP76-F*	ATACGCTTAGAGGTGGTGGAA
*SfJHBP76-R*	TCTGATGAGGTTGGTGAGGTT

## Data Availability

The data presented in this study are available upon request from the corresponding author.

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
