# Peer review of "Genome-Wide Identification and Analysis of Family Members with Juvenile Hormone Binding Protein Domains in Spodoptera frugiperda"

_insects, 2024, doi:10.3390/insects15080573_

Round 1

Reviewer 1 Report

Comments and Suggestions for Authors

This MS entitled “Genome-wide identification and expression analysis of the JHBP gene family in Spodoptera frugiperda” by Liu and colleagues extracted JHBP related sequences from the genomic DNA sequence. The transcriptome data which have been uploaded at the NCBI site were used for expression analysis of JHBPs. Some of them were analyzed by q-PCR. The linage of JHBP related sequences were estimated by bioinformatics.

 The MS contains some problems for publication.

A major concern,

The dataset which used for extraction of JHBP related sequences contains many takeout genes. JHBPs are one of group of takeout family. There is a possibility that the JHBPs mentioned in the MS involve takeout genes which are not true JHBPs. It is better to distinguish between JHBP and takeout.

Minor concerns,

1.     The expression analysis in cuticle were presented in figures 7 and 9. Cuticles does not contain cells. Did you use Integument or epidermis?

2.     All figures are too small. Please enlarge the figures as optimal sizes.

3.     The mRNA expression levels of 13 JHBP related genes were measured by Q-PCR. I wondered why the authors selected these genes. In the MS, there is not any explanation.

4.     Reference lists were not written correctly. 

Reviewer 2 Report

Comments and Suggestions for Authors

Authors present genome-wide identification and expression analysis for Juvenile hormone binding proteins in the fall army worm (Spodoptera frugiperda). The idea to re-use public RNA sequencing data in NCBI GEO is good point and the analysis pipeline for these seems to be fine, but the descriptions about the pipeline must be revised to be published as so many important information are missing in current version of then manuscript.

Major issues

  1. Abbreviations:
    1. Abbreviation should not be used in the title. ‘JHBP’ should be ‘Juvenile hormone binding protein’.
    2. So many abbreviations are re-defined in the manuscript again and again. For example, JHBP was first defined in line 57, but it was also defined in line 64. Authors should check this type of incompleteness of the manuscript in the revision.
  2. The description about why JHBP was targeted in the study seems inadequate. More description for that should be included in the Introduction section.
  3. The version information about tools used in the analysis is completely missing in the Materials and Methods section. For example, the use of Pfam database was described in line 96, but there is no description about the Pfam version. The version must be described for the biological database used. If the database information cannot be found in the tools, authors should describe the date of the searches.This issue should be applicable for all database and tool in the manuscript.
  4. In line 100, authors described “Pfam (http://pfam.wust.ledu/hmmsearch.shtml)”, but Pfam is the name of database, not the tool for searching Pfam with specific HMM model. It should be “HMMER”. Authors must cite the paper for HMMER also.
  5. Figures: most figures (Figures 1, 4, and 5) in the manuscript were too small
    1. Reviewer recognizes that Figure 1 is very important figure in the manuscript, but is is not sufficient size. Authors should enlarge this figure for the visibility.
    2. Reviewer cannot see Figure 2 as it is too small. This figure must be rearranged for its visibility. 
  6. The availability of the data: the important table for 76JHBP genes are listed in the supplemental data. It should be described in ‘Data Availability Statement’ section. 

Minor issues

  1. In line 85-86, authors described “The research analyzed the potential role of the gene and validated its function for future gene knockout experiments and the application of RNAi technology”, but the RNAi is not ‘knockout’, but ‘knockdown’
  2. The nomenclature of species should be rechecked. For example, silkworm should be ‘Bombyx mori’ not ‘Bombyxmori’ (in Line 339 and 344) and ‘Bombyx mori Linnaeus’ (Line 391).
Comments on the Quality of English Language

Some expression seems to be questionable. Moderate English editing ris equired.

Round 2

Reviewer 1 Report

Comments and Suggestions for Authors

In the revised manuscript, authors have addressed all of my concerns by either through clarification or rewrote except for the question for “cuticle”. I mentioned in the review report for 1st MS that cuticle does not contain cells. The authors still used this term as entire epidermis. I recommend that “cuticle” is replaced with “integument” or “entire epidermis”.

Some minor points are listed below,

Line 182, 351, 353, 355, 364, 457, and in Fig.8, Please replace all “cuticle” with “integument” or “entire epidermis”.

Ref 29 and 58, the journals were not showed in the list.

Ref 30, the title is shown in abbreviated form.

Author Response

请参阅附件。

Reviewer 2 Report

Comments and Suggestions for Authors

Nearly all issues the reviewer raised in the previous round were resolved, but one major issue (Comment 8) was not revised.

> Comments 8: The availability of the data: the important table for 76JHBP genes are listed in the supplemental data. It should be described in ‘Data Availability Statement’ section.

> Response 8: We agree with the suggestion and express our gratitude. We have included a description of the important table for 76 JHBP genes in the 'Data Availability Statement' section. Thank you for your valuable feedback.

'Data Availability Statement' was not updated in the manuscript file while the authors described above in the rebuttal letter.
